# Can the Control of Chronic Spontaneous Urticaria Symptoms Depend on the Stress-Coping Styles?

**DOI:** 10.3390/diagnostics15232965

**Published:** 2025-11-22

**Authors:** Marzena Pluta-Kubicz, Edyta Jura-Szołtys, Radosław Gawlik, Magdalena Feusette, Robert Okuniewicz, Zenon Brzoza

**Affiliations:** 1Department of Internal Diseases, Allergology and Clinical Immunology, Medical University of Silesia, 40-752 Katowice, Poland; marzenakubicz@wp.pl (M.P.-K.); rgawlik@sum.edu.pl (R.G.); 2Department of Internal Diseases, Allergology, Endocrinology and Gastroenterology, Institute of Medical Sciences, University of Opole, 45-401 Opole, Poland; magdalena.feusette@uni.opole.pl (M.F.); robert.okuniewicz@uni.opole.pl (R.O.); zbrzoza@mp.pl (Z.B.)

**Keywords:** chronic urticaria, Urticaria Control Test, Coping Inventory for Stressful Situations questionnaire

## Abstract

**Background:** The symptoms of chronic spontaneous urticaria can be exacerbated or even induced by psychological stress. Assessing the severity of symptoms using the recommended Urticaria Control Test is an important diagnostic step before deciding on the type of pharmacological treatment to be used. Due to the possibility of urticaria symptoms affecting patient’s emotional condition, the authors attempted to analyze if the way of coping with stress has an impact on urticaria symptom control as assessed with this questionnaire. **Methods:** The study included 61 (37 female; 60,6%;) patients with symptoms of chronic spontaneous urticaria without other coexisting diseases. All patients were treated with antihistamines. In the analyzed group of patients, the Urticaria Control Test and the Polish version of Endler and Parker’s Coping Inventory for Stressful Situations questionnaire were conducted. **Results:** The average score on the Urticaria Control Test in the analyzed group was 8.5 (±3.9) points. In our group, the most common coping style was a mixed style based on emotions and avoidance—32 (53%) respondents. Next, 11 (18%) patients reported an emotion-based style. A task-oriented style of coping with stress was observed in 8 (13%) respondents. In the study group, we found no statistical significance in the correlation between the UCT results and the patient’s coping style. **Conclusions:** Emotions play a significant role as a stress-coping style in chronic spontaneous urticaria patients. The lack of relation found between the Urticaria Control Test result and the Coping Inventory for Stressful Situations questionnaire confirms the objective usefulness of the Urticaria Control Test in assessing the control of spontaneous urticaria.

## 1. Introduction

Chronic urticaria affects approximately 0.5–1% of the general population [1]. It can be either spontaneous or inducible [2]. Chronic spontaneous urticaria (CSU) is a skin disease characterized by recurrent hives, itching, and/or angioedema that recur for more than six weeks. Symptoms appear without any apparent cause [3]. The symptoms of urticaria can be exacerbated or even induced by psychological stress. In clinical practice, a significant proportion of patients with symptoms of chronic urticaria have been found to have experienced a major stressful event in the period preceding the onset of the disease. Research has confirmed that the skin functions not only as a physical barrier but also as a dynamic sensory organ that can directly perceive stress signals. It responds to psychological and physiological stress through complex neuroendocrine and immune mechanisms, making it both an immediate perceiver of stress and a target of stress-induced reactions such as inflammation and itching. Psychological stress activates the neuroendocrine system, leading to increased secretion of cortisol and catecholamines (adrenaline, noradrenaline). These hormones affect mast cells, which play a crucial role in the pathogenesis of urticaria, leading to their degranulation and the release of histamine—the main mediator of urticaria symptoms. These mediators contribute to pruritus and wheal formation, which in turn intensify emotional distress. The resulting scratching and sleep disturbances further aggravate inflammation and symptom severity. This interplay creates a self-perpetuating vicious cycle—stress exacerbates urticaria, and worsening symptoms heighten stress—highlighting the importance of a holistic therapeutic approach. Stress exacerbates the symptoms of urticaria, and the symptoms worsen psychological conditions, intensifying stress. The relationship between stress and urticaria is complex and multifaceted [4]. The pathogenesis of the impact of stress factors on the development of a vicious cycle in the course of chronic urticaria is presented in Figure 1.

The symptoms of chronic urticaria, such as itching, skin lesions, and unpredictability of symptom occurrence, significantly reduce patients’ quality of life. They can lead to psychological problems, low self-esteem, and social isolation. They have a negative impact on family and professional life [5,6]. A significant deterioration in quality of life across multiple domains, including social interactions, professional work, sleep, and emotional relationships, was observed in patients with chronic urticaria who additionally reported symptoms of pressure-induced delayed urticaria [7]. Previous studies have shown that the quality of life of patients with chronic urticaria is comparable to that of people suffering from severe asthma or type 1 diabetes [8]. A two-year study conducted in Italy demonstrated that CSU imposes a considerable socio-economic burden, and that improvement in quality of life can be achieved through the implementation of an appropriate therapeutic pathway [9]. Laboratory tests in this group of patients have limited diagnostic value because they rarely indicate the cause of symptoms. Therefore, assessing the severity of symptoms using the recommended questionnaire is an important diagnostic step before deciding on the modification of pharmacological treatment to be used [2]. The recommended Urticaria Control Test (UCT) questionnaire is used to assess the degree of control of urticaria symptoms. Recommended by international guidelines (EAACI/GA^2^LEN/EDF/WAO), the UCT not only allows for the assessment of symptom control, but also is a tool for evaluating treatment progress and making therapeutic decisions [10].

Due to the possibility of urticaria symptoms affecting patient’s emotional condition, the authors attempted to analyze if the way of coping with stress has an impact on urticaria symptom control as assessed with the UCT.

## 2. Materials and Methods

### 2.1. Study Design and Participants

Participation in the study was completely voluntary. Participants were informed about the confidentiality of the collected data. In this real-life study, we analyzed UCT results in adult CSU patients. All patients also underwent the questionnaire diagnosing stress-coping styles (CISS) [11]. Data of consecutive patients admitted to the Department were analyzed.

### 2.2. Inclusion Criteria

Inclusion criteria were as follows: age equal or over 18 years, symptoms of CSU diagnosed according to the EAACI/GA^2^LEN/EuroGuiDerm/APAAACI international criteria [3].

### 2.3. Exclusion Criteria

Exclusion criteria were as follows: patients under 18 years of age, pregnancy, other coexisting severe diseases, the use of systemic steroids less than 4 weeks prior to the examination.

### 2.4. Urticaria Control Test (UCT)

The UCT questionnaire consists of four questions that the patient assesses independently, referring to the last 4 weeks. For each question, the patient can score between 0 and 4 points, where 0 is the worst score and 4 is the best. The scores are added up. A score below 12 points indicates a lack of control over urticaria and requires more intensive treatment or consideration of other therapeutic options. A score of 12–15 points indicates good control of urticaria, and possibility to continue and optimize the current treatment. A score of 16 points indicates complete control of urticaria symptoms and the possibility of modifying the current pharmacotherapy.

### 2.5. Coping Inventory for Stressful Situations (CISS)

In order to answer the research question, the Polish version of Endler and Parker’s Coping Inventory for Stressful Situations (CISS) was used. The adult version of the CISS is suitable for individuals who are 18 years of age and older.

The CISS measures three main styles of coping with stress:-Task-Oriented Coping (taking action to achieve the objective);-Emotion-Oriented Coping (activities that reduce emotions by focusing on oneself, wishful thinking);-Avoidance-Oriented Coping (avoiding thinking about the difficult situation).

The avoidance-focused style takes two forms: engaging in substitute activities or seeking social contact.

The CISS test contains 48 items (statements). Responses are given on a 5-point Likert scale (from “never” to “very often”). Next to each statement are numbers from 1 to 5, which determine the frequency with which a given activity is undertaken by the respondent in stressful situations. Respondents should respond to each statement by circling the appropriate number. Respondents can score between 16 and 80 points. The number of items included in each scale is specified in a separate code. The values obtained after summing each scale are called the raw research data. In the next phase, the results of individual patients were compared to the “sten” norms listed in the appropriate table in the test annex. Then, in accordance with the formula recommended in the test, the “sten” results for individual scales were converted to obtain information about the dominant style of coping with stress used. The test takes about 10–15 min to complete.

### 2.6. Statistical Analysis

In statistical analyses, we used Spearman and Chi-square tests. *p* values less than 0.05 were considered statistically significant (Statistica 13.3, StatSoft GmbH, Hamburg, Germany).

We aimed to compare in our patients the prevalence of different stress-coping styles.

## 3. Results

Sixty-one patients (37 female, 60,6%; median and IQR of age 44, 25–70) with CSU without coexisting diseases were qualified for the assessment. All patients were on antihistamine therapy and completed the UCT. The average test score in the analyzed group was 8.5 (±3.9) points. Each patient also completed the CISS questionnaire.

In the analyzed group, the most statistically significant common coping style, in comparison to others, was a mixed style based on emotions and avoidance, which was reported by 32 (53%) respondents. Next, 11 (18%) patients reported an emotion-based style. A task-oriented style of coping with stress was observed in eight (13%) respondents. In a group of five (8%) patients, due to discrepancies in the results, it was not possible to determine the style of coping with stress. Three (5%) respondents represented a mixed task- and emotion-based style, while two (3%) represented an avoidance-based style. Due to the small group of patients using an avoidance-based stress-coping style, no values were assigned in the subscales for engaging in substitute activities and seeking social contact. The percentage distribution and statistical significance of stress-coping styles in the study group are shown in Figure 2 and in Table 1.

In the study group, we found no statistical significance in the correlation between the UCT results and the patient’s coping style (Table 2).

## 4. Discussion

The final diagnosis of CSU is established after the diagnostic procedures of inducible causes of chronic urticaria have been excluded. It is also clinically important to exclude urticaria-like disorders in differential diagnostics. In some patients, the disease resolves spontaneously within 1–5 years, but in others it requires long-term control with medication [2]. Chronic urticaria generates stress, and stress exacerbates the symptoms of the disease. Stress and emotions can exacerbate the symptoms of the disease [12]. Cortisol and adrenaline affect mast cells and can increase itching or the frequency of hives [13]. In the treatment of CSU, the recommended antihistamines are often insufficient. Currently, anti-IgE biological treatment with the effective and safe omalizumab is recommended for this group of patients. In order to qualify for biological treatment, the objective severity of urticaria symptoms is required [14].

The UCT is recommended for the diagnosis and monitoring of urticaria symptoms and can support therapeutic decisions. It is a standardized, validated self-assessment tool used to evaluate the degree of control of disease symptoms by the patient over the past 4 weeks. It has proven accuracy, reliability, and ability to detect clinical change [15,16]. It is available in 30 languages, including Polish [17]. The UCT is also a useful tool for patient education. It allows patients to consciously monitor their symptoms, understand the impact of treatment on their daily functioning, and participate in therapeutic decision-making, known as shared decision-making. The UCT plays an important role in the “treat-to-target” (TTT) approach. Modern management of chronic diseases such as urticaria is based on the concept of “treating to target”, i.e., setting a specific clinical goal and adjusting treatment until that goal is achieved [18,19]. Because of the well-documented impact of stress and emotions on urticaria symptoms, the authors attempted to assess whether a patient’s coping style could influence the results of the UCT. In order to assess stress coping, the authors conducted a CISS test in a study group of 61 patients with CSU. CISS has been characterized by good psychometric properties and is recommended as an adequate and reliable tool for assessing coping styles [20]. The CISS test results of individuals who demonstrate difficulties in managing stress and in taking effective interventions to prevent stress reactions correlate positively with emotional and avoidant styles [21]. The task-oriented style of coping with stress was confirmed as dominant in a group of professional soldiers and physicians who effectively target the management of stressful situations [22,23]. It should be emphasized that the CISS test allows for the analysis of individual predispositions in coping with stressful situations, whereas the stress factor itself is not the subject of this analysis. The analysis of the results in our group of patients was conducted in accordance with the stages recommended by the authors of the test. Analysis of the results obtained allowed us to identify the dominant way of coping with stress in the group of patients with CSU. The analysis conducted by the authors showed the highest percentage and a statistically significant higher frequency of occurrence of mixed coping styles focused on emotions and avoidance in the total group of respondents. Next was the emotion-focused style. In the authors’ opinion, the analysis of coping styles confirms the role of emotions in the etiology of chronic spontaneous urticaria. In addition to targeted pharmacotherapy, the results obtained indicate the need for potent cooperation between psychologists and psychiatrists in controlling the symptoms of chronic spontaneous urticaria.

In response to the main research question, no statistical significance was found between the Urticaria Control Test result and the patient’s coping with stress, which confirms the objective usefulness of the UCT in assessing the control of CSU. The lack of a significant correlation between the severity of urticaria symptoms and coping styles, as assessed using the CISS questionnaire, may indicate that the intensity of experienced stress, rather than the preferred coping strategies, plays a more critical role in the clinical expression of the disease. It is important to note that the coping style represents a relatively stable personality trait, whereas stress intensity is a dynamic and context-dependent variable influenced by current psychosocial conditions. Therefore, it is plausible that, within the studied cohort, variations in stress intensity exerted a greater impact on disease activity than individual differences in coping patterns. Nevertheless, an indirect effect of the coping style cannot be entirely ruled out, for example, through its influence on the subjective appraisal of stress or on the effectiveness of emotion regulation processes. Further studies incorporating both stress intensity and subjective stress appraisal are warranted to elucidate the complex interplay between psychological factors and disease activity in chronic urticaria. No similar studies on urticaria were found in the available literature that would allow us to compare the results of our study with the reports of other authors.

## 5. Conclusions

Based on the analysis of the obtained results, it was confirmed that emotions play a significant role as a stress-coping style in patients with chronic spontaneous urticaria.

Our analysis of stress-coping styles in this group of patients also opens new perspectives for further research aimed at developing novel therapeutic approaches in collaboration with psychologists and psychiatrists. 

Furthermore, the absence of a relationship between the Urticaria Control Test results and the Coping Inventory for Stressful Situations questionnaire confirmed the objective usefulness of the Urticaria Control Test in assessing the level of disease control in patients with chronic spontaneous urticaria.

## Figures and Tables

**Figure 1 diagnostics-15-02965-f001:**
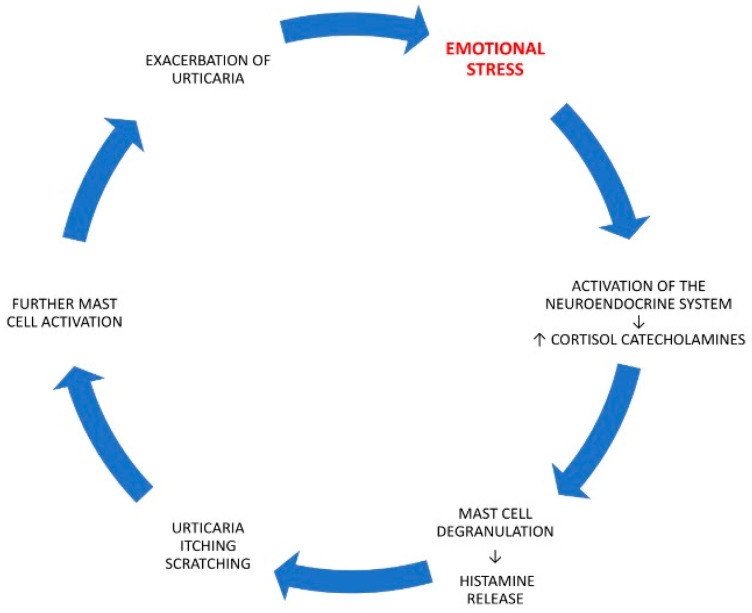
The pathogenesis of the impact of stress factors on the development of a vicious cycle in the course of chronic urticaria. Stress, as the factor initiating the vicious-cycle process in the course of chronic spontaneous urticaria, has been highlighted in red. The meaning of the arrows in the figure has been explained in the text.

**Figure 2 diagnostics-15-02965-f002:**
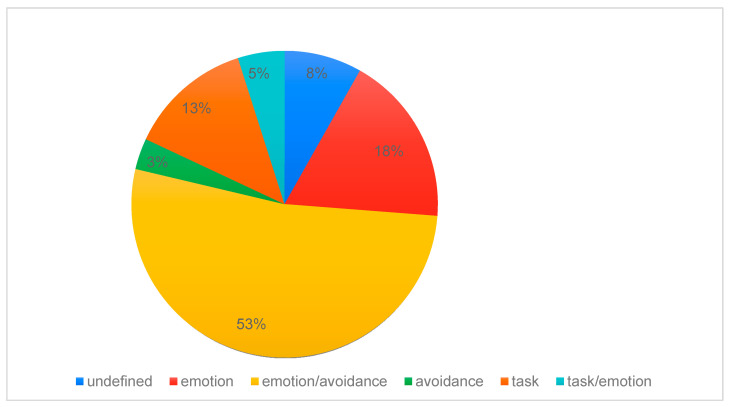
Percentage distribution of stress-coping styles in the chronic spontaneous urticaria patients.

**Table 1 diagnostics-15-02965-t001:** Frequency of styles of coping with stress comparison in chronic spontaneous urticaria patients (CISS—Coping Inventory for Stressful Situations, *p*-statistical significance of Chi-square test).

CISS	
Emotion/avoidance vs. Emotions	*p* = 0.01
Emotions/avoidance vs. Task	*p* < 0.01
Emotions/avoidance vs. Undefined	*p* < 0.01
Emotion/avoidance vs. Task/emotion	*p* < 0.01
Emotion/avoidance vs. Avoidance	*p* < 0.01
Emotion vs. Task	*p* = 0.03
Emotion vs. Undefined	*p* < 0.01
Emotion vs. Task/emotion	*p* < 0.01
Emotion vs. Avoidance	*p* < 0.01
Task vs. Undefined	*p* = 0.38
Task vs. Task/emotion	*p* = 0.11
Task vs. Avoidance	*p* = 0.04
Undefined vs. Task/emotion	*p* = 0.47
Undefined vs. Avoidance	*p* = 0.24
Task/emotion vs. Avoidance	*p* = 0.65

**Table 2 diagnostics-15-02965-t002:** Demographic characteristics of patients and the results of the tests performed (UCT—Urticaria Control Test, CISS-T—Coping Inventory for Stressful Situations Task, CISS-E—Coping Inventory for Stressful Situations Emotion, CISS-A—Coping Inventory for Stressful Situations Avoidance, *p*-statistical significance of Spearman test).

Age (Years; Mean ± SD)	61 ± 14.2	
Male *n* (%)	24 (39.4)	
Female *n* (%)	37 (60.6)	
UCT	8.5 (±3.9)	
CISS-T	52.6 (36–68)	
CISS-E	45.2 (33–64)	
CISS-A	45.1 (25–61)	
UCT and CISS-T	*p* = 0.56	R = 0.07
UCT and CISS-E	*p* = 0.77	R = −0.04
UCT and CISS-A	*p* = 0.17	R = 0.18

## Data Availability

All data generated or analyzed during this study are included in this article. Further enquiries can be directed to the corresponding author.

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
