# Peer review of "Can the Control of Chronic Spontaneous Urticaria Symptoms Depend on the Stress-Coping Styles?"

_diagnostics, 2025, doi:10.3390/diagnostics15232965_

Round 1

Reviewer 1 Report

Comments and Suggestions for Authors

The authors of the manuscript address a highly relevant issue of the dependence of urticaria activity on stress factors. The burden of chronic spontaneous urticaria is associated with a substantial disease burden, which is confirmed by the results of numerous studies on quality of life.

Previous researches have explored the relationship between the severity of an emotional triggers and urticaria`s exacerbations. However, the investigators focus on a previously unstudied question: the potential dependence of urticaria severity on an individual's behavior in a stressful situation and their coping style.

The results of the conducted study did not reveal a significant correlation between the severity of urticaria, as objectively assessed by the validated UCT, and the style of coping with stress according to the CISS. This suggests that the crucial factor is the intensity of the stress itself, rather than the methods of coping with it.

It would be wonderful to add a figure or diagram illustrating the pathogenesis of the influence of stress factors on mast cells with the release of pro-inflammatory mediators. This would allow for better visualization of this important point. One could also elaborate slightly more on the formation of a vicious cycle: emotional stress → exacerbation of urticaria → stress, itching, skin scratching → further exacerbation of urticaria.

I would recommend adding a few more research findings to emphasize the impact of chronic urticaria on patients' quality of life. The authors may wish to consider the suggested references to support this point:

  • O'Donnell BF, Lawlor F, Simpson J, et al. The impact of chronic urticaria on the quality of life. Br J Dermatol 1997; 136:197.
  • Rossi O, Piccirillo A, Iemoli E, et al. Socio-economic burden and resource utilisation in Italian patients with chronic urticaria: 2-year data from the AWARE study. World Allergy Organ J 2020; 13:100470.

The authors have conducted interesting and very important research, and I hope they will continue to study this crucial aspect of the relationship between urticaria and emotional behavioral responses.

Author Response

The authors of the manuscript address a highly relevant issue of the dependence of urticaria activity on stress factors. The burden of chronic spontaneous urticaria is associated with a substantial disease burden, which is confirmed by the results of numerous studies on quality of life. Previous researches have explored the relationship between the severity of an emotional triggers and urticaria`s exacerbations. However, the investigators focus on a previously unstudied question: the potential dependence of urticaria severity on an individual's behavior in a stressful situation and their coping style.

We would like to sincerely thank you for taking the time to carefully review our manuscript and for providing valuable comments and suggestions. Below, we present detailed responses to all reviewers’ remarks. The corresponding revisions have been incorporated into the manuscript and are highlighted in the re-submitted files to facilitate their identification

1.The results of the conducted study did not reveal a significant correlation between the severity of urticaria, as objectively assessed by the validated UCT, and the style of coping with stress according to the CISS. This suggests that the crucial factor is the intensity of the stress itself, rather than the methods of coping with it.

Thank you for drawing attention to this aspect of the interpretation of the results. We agree that the lack of a significant correlation between the severity of urticaria and coping styles (as measured by the CISS) may suggest that it is not the manner of coping with stress, but rather the intensity of the stress itself, that plays a more important role in the course of the disease.
It is important to note that coping style represents a relatively stable personality trait, whereas stress intensity is a dynamic and context-dependent variable influenced by current psychosocial conditions. Therefore, it is plausible that, within the studied cohort, variations in stress intensity exerted a greater impact on disease activity than individual differences in coping patterns. Nevertheless, an indirect effect of coping style cannot be entirely ruled out, for example through its influence on the subjective appraisal of stress or on the effectiveness of emotion regulation processes. Further studies incorporating both stress intensity and subjective stress appraisal are warranted to elucidate the complex interplay between psychological factors and disease activity in chronic urticaria.
This aspect of the analysis of results was included in the discussion.

2.It would be wonderful to add a figure or diagram illustrating the pathogenesis of the influence of stress factors on mast cells with the release of pro-inflammatory mediators. This would allow for better visualization of this important point. One could also elaborate slightly more on the formation of a vicious cycle: emotional stress → exacerbation of urticaria → stress, itching, skin scratching → further exacerbation of urticaria.

Thank you for this suggestion. A diagram illustrating the pathogenesis of the influence of stress factors on mast cells and the release of pro-inflammatory mediators has been added to the Introduction. The description of the formation of the vicious cycle initiated by the stress response, which leads to the exacerbation of urticaria symptoms, has also been expanded.

3.I would recommend adding a few more research findings to emphasize the impact of chronic urticaria on patients' quality of life. The authors may wish to consider the suggested references to support this point:

  • O'Donnell BF, Lawlor F, Simpson J, et al. The impact of chronic urticaria on the quality of life. Br J Dermatol 1997; 136:197.
  • Rossi O, Piccirillo A, Iemoli E, et al. Socio-economic burden and resource utilisation in Italian patients with chronic urticaria: 2-year data from the AWARE study. World Allergy Organ J 2020; 13:100470.

We would also like to thank you for suggesting additional references, which have been carefully reviewed and incorporated into the manuscript as they significantly enrich and strengthen the context of our work.

The authors have conducted interesting and very important research, and I hope they will continue to study this crucial aspect of the relationship between urticaria and emotional behavioral responses.

Once again, we sincerely thank you for your valuable review of our manuscript. We are currently continuing our observations as part of the ongoing study, which we hope will provide additional data and further insights in the future.
We kindly extend our respectful regards on behalf of all the authors.

Reviewer 2 Report

Comments and Suggestions for Authors

an interesting study by Pluta-Kubicz et al regarding an "unexplored" field that clinicians always think about but have no idea of how to approach or deal with it.

the thing is that part of the references are irrelevant, cause i can humbly confess that i don't now if the authors may compare a stress related to CSU with that of army or medical environment related stress, so i recommend the authors to clarify on this point, or to delete the references regarding those

Author Response

an interesting study by Pluta-Kubicz et al regarding an "unexplored" field that clinicians always think about but have no idea of how to approach or deal with it.

We sincerely thank you for the time you devoted to reviewing our manuscript.

the thing is that part of the references are irrelevant, cause i can humbly confess that i don't now if the authors may compare a stress related to CSU with that of army or medical environment related stress, so i recommend the authors to clarify on this point, or to delete the references regarding those

We appreciate the valuable suggestion. We agree that providing further clarification on this issue will enhance the clarity of the text. The authors conducted an analysis of coping strategies in a group of patients with chronic urticaria and compared the obtained results with those from other professional groups. These findings confirm the significant role of the emotional factor in the pathogenesis of chronic urticaria. In contrast, the results of the CISS test in the group of professional soldiers and physicians indicated a predominance of task-oriented coping style, suggesting a stable emotional orientation in these individuals. It should be emphasized that the CISS test allows for the analysis of individual predispositions in coping with stressful situations, whereas the stress factor itself is not the subject of this analysis.

The information has been provided in the manuscript and are highlighted in the resubmitted files to facilitate their identification.
We kindly extend our respectful regards on behalf of all the authors